# The Immunosuppressive Niche Established with a Curcumin-Loaded Electrospun Nanofibrous Membrane Promotes Cartilage Regeneration in Immunocompetent Animals

**DOI:** 10.3390/membranes13030335

**Published:** 2023-03-14

**Authors:** Yu Zhang, Renzhong Cai, Jun Li, Xu Wu

**Affiliations:** 1Department of Thoracic and Cardiovascular Surgery/Huiqiao Medical Center, Nanfang Hospital, Southern Medical University, Guangzhou 510515, China; 2Department of Breast Surgery, Hainan General Hospital, Hainan Hospital Affiliated to Hainan Medical College, Haikou 570311, China; 3Department of Thoracic Surgery, Hainan General Hospital, Hainan Hospital Affiliated to Hainan Medical College, Haikou 570311, China

**Keywords:** immunosuppressive niche, curcumin, cartilage regeneration, immunocompetent animal, electrospun membrane

## Abstract

Inflammatory cells mount an immune response against in vitro engineered cartilage implanted into immunocompetent animals, consequently limiting the usage of tissue-engineered cartilage to repair cartilage defects. In this study, curcumin (Cur)—an anti-inflammatory agent—was mixed with poly(lactic-co-glycolic acid) (PLGA) to develop a Cur/PLGA nanofibrous membrane with nanoscale pore size and anti-inflammatory properties. Fourier-transform infrared spectroscopy and high-performance liquid chromatography analyses confirmed the successful loading of Cur into the Cur/PLGA nanofibrous membrane. The results of the in vitro assay demonstrated the sustained release kinetics and enhanced stability of Cur in the Cur/PLGA nanofibrous membrane. Western blotting and enzyme-linked immunosorbent assay analyses revealed that the Cur/PLGA nanofibrous membrane significantly downregulated the expression of inflammatory cytokines (IL-1β, IL-6, and TNF-α). A chondrocyte suspension was seeded into a porous PLGA scaffold, and the loaded scaffold was cultured for 3 weeks in vitro to engineer cartilage tissues. The cartilage was packed with the in vitro engineered Cur/PLGA nanofibrous membrane and subcutaneously implanted into rats to generate an immunosuppressive niche. Compared with those in the PLGA-implanted and pure cartilage (without nanofibrous membrane package)-implanted groups, the cartilage was well preserved and the inflammatory response was suppressed in the Cur/PLGA-implanted group at weeks 2 and 4 post-implantation. Thus, this study demonstrated that packaging the cartilage with the Cur/PLGA nanofibrous membrane effectively generated an immunosuppressive niche to protect the cartilage against inflammatory invasion. These findings enable the clinical translation of tissue-engineered cartilage to repair cartilage defects.

## 1. Introduction

Cartilage defects resulting from trauma, tumors, and inflammation are commonly encountered in clinics [1]. The self-repair capacity of cartilage tissues is limited, as they lack the vascular, lymphatic, and nervous systems. Even minor injuries may lead to progressive damage to cartilage [2]. Currently, practical treatment for cartilage defects is not available [3]. Tissue engineering using the combination of cells, bioactive factors, and functional biomaterials is a promising approach to repair damaged cartilage [4].

Globally, several researchers have aimed to mitigate cartilage defects using cartilage tissue engineering technology. Zhou and Cao et al. reported the first application of an in vitro regenerated ear-shaped cartilage for auricular reconstruction [5]. However, the authors reported a 60% success rate for ear restoration with predetermined cartilage quality. The other 40% of the engineered cartilages were adversely affected by inflammatory cell infiltration, resulting in the erosion and absorption of the cartilage’s extracellular matrix (ECM). Other studies have also reported immune-cell-mediated cartilage destruction [6,7,8]. Hence, there is a need to develop strategies to protect the tissue-engineered cartilage against inflammatory cell infiltration.

Several strategies have been proposed to mitigate the immune reaction against in vitro engineered cartilage after in vivo implantation. Shen et al. developed electrospun acid-neutralizing fibers to ameliorate inflammatory responses [9]. Wang et al. [10] and Jia et al. [11] proposed a decellularization method to alleviate the inflammatory response to facilitate cartilage regeneration in immunocompetent large animals. Li et al. proposed the use of a cell sheet without any scaffold to decrease inflammatory responses and promote submuscular cartilage formation [12]. However, these methods are not effective to mitigate cell-induced and surgery-induced immune reactions. Liu et al. suggested that prolonged in vitro pre-cultivation could alleviate post-implantation inflammation and promote stable subcutaneous cartilage formation in a goat model [13], but the prolonged cultivation period increased the time and cost of the procedure. Yang et al. introduced a drug-loaded porous scaffold with anti-inflammatory activities to enhance tracheal cartilage regeneration [14]. However, the direct contact of in vitro engineered cartilage with the immunological system promoted the erosion and absorption of the cartilage’s ECM. Gao et al. developed a skin-derived epithelial lining to facilitate tracheal transplantation by protecting the tracheal cartilage against inflammatory erosion [15]. However, the usage of autologous tissue resulted in donor site damage. Hence, there is an urgent need to develop a simple and effective strategy to protect the in vitro engineered cartilage against inflammatory responses in an in vivo environment.

Recently, a novel immunoisolation strategy using semipermeable membrane materials has emerged as a feasible solution to protect tissue from immune attack by the host [16]. For cartilage regeneration, it is essential to choose a proper semipermeable membrane for simultaneous efficient immunoisolation and stable chondrocyte survival. A nanofibrous membrane synthesized via an electrospinning process is a type of fishing-net-like vehicle with nanoscale pore size [17], representing a logically suitable semipermeable membrane for immunoisolation and nutrition exchange. Additionally, the nanofibrous membrane is suitable for drug loading owing to its ultrahigh specific surface area [18]. Curcumin (Cur), a natural flavone, exerts potent therapeutic effects on acute and chronic inflammation [19,20]. Moreover, compared with other anti-inflammatory drugs, Cur is associated with several advantages in terms of broad resources and cost-effectiveness [21]. In this study, Cur was loaded into the routinely used poly(lactic-co-glycolic acid) (PLGA). This mixture was subjected to an electrospinning procedure to develop a Cur/PLGA nanofibrous membrane with anti-inflammatory properties. Next, the prepared Cur/PLGA nanofibrous membrane was used to tightly wrap the in vitro engineered cartilage. The nanofiber may physically obstruct the infiltration of inflammatory cells, while the released Cur may suppress inflammation-related cytokines, generating an immunosuppressive niche to protect the in vitro engineered cartilage against inflammatory response after in vivo implantation. Thus, this strategy aims to promote cartilage regeneration in immunocompetent animals.

## 2. Materials and Methods

### 2.1. Synthesis of the Cur/PLGA Nanofibrous Membrane

PLGA (molecular weight = 250,000, L/G 75/25, Shanghai Yuanye Biotechnology Co., Ltd., Shanghai, China) was dissolved in trichloromethane solution at a concentration of 10% (*w*/*v*) for the fabrication of the PLGA nanofibrous membrane using a conventional electrospinning method [22]. Additionally, Cur and PLGA were mixed at a ratio of 1:5 (*w*/*w*), and the mixture was dissolved in trichloromethane solution at a concentration of 10% (*w*/*v*) for the fabrication of the Cur/PLGA nanofibrous membrane. The obtained PLGA and Cur/PLGA nanofibrous membranes were disinfected using the ultraviolet (UV) irradiation method.

### 2.2. Characterization of the Cur/PLGA Nanofibrous Membrane

The macromorphologies of the PLGA and Cur/PLGA nanofibrous membranes were examined using an SLR camera (Nikon, Tokyo, Japan), while the micromorphologies were assessed using a scanning electron microscope (Hitachi TM-1000, Tokyo, Japan). The samples were rinsed with PBS and fixed overnight in 0.05% glutaraldehyde at 4 °C. After dehydration through a graded series of ethanol, the samples were critical-point dried and sputtered using platinum for 3 min. Thereafter, the nanofibrous membranes were observed at an accelerating voltage of 15 kV. The porosity of the PLGA and Cur/PLGA nanofibrous membranes was calculated from the obtained images using Photoshop 2021 software.

The Fourier-transform infrared (FTIR) spectra of the PLGA and Cur/PLGA nanofibrous membranes were recorded using a Varian model 640-IR spectrometer (Varian Inc., Palo Alto, CA, USA). The spectra were recorded under the following conditions using Varian Resolution software (Varian Inc.): wavelength, 750–4000 cm^−1^; resolution, 4.0 cm^−1^; scanning, 64×.

High-performance liquid chromatography (HPLC) was performed to analyze the samples in the Cur, PLGA, and Cur/PLGA groups. The conditions for HPLC were as follows: chromatograph, HPLC system (Agilent 1100 series, Waldbronn, Germany); column, Zorbax Eclipse C18 column (250 × 4.0 mm^2^); mobile phase, 1% acetic acid in water and 52% acetonitrile; temperature, room temperature; flow rate, 1 mL/min; injection volume, 5 μL; detection wavelength, 424 nm. The column was equilibrated with the mobile phase for 10 min and washed with 100% acetonitrile for 10 min.

UV–visible spectrophotometry was performed to determine the in vitro release kinetics of Cur from the Cur/PLGA membrane. The membrane (20 mg) was incubated in 0.1 M phosphate-buffered saline (PBS, pH 7.4) in a 100 mL flask that was hermetically sealed and protected from light, stirred at 100 rpm, and incubated in a shaking water bath at 37 °C. Aliquots of samples (5 mL) were periodically removed from the flask to measure the cumulative amount of Cur released. The measurements were performed using a Shimadzu UV-1603 dual-beam UV spectrophotometer (Japan) at an absorption maximum of 421 nm.

Both Cur and Cur/PLGA samples with equal amounts of Cur (marked as α) were soaked in PBS (pH 7.4) at 37 °C to observe their stability. An aliquot (200 μL) of the solution in each group was periodically removed over 1–8 h to determine the soluble Cur content (marked as β). The remaining Cur content was calculated as follows: (α − β)/α × 100%.

### 2.3. Chondrocyte and RAW264.7 Macrophage Culturing

All animal experiments were approved by the Animal Care and Use Committee of Nanfang Hospital. Chondrocytes were isolated from the articular cartilage of Sprague Dawley rats (aged 6–8 weeks, Shanghai SLAC Laboratory Animal Co., Ltd., Shanghai, China), following a previously established method [23]. Briefly, articular cartilage tissues were harvested from rats and digested with 0.25% type II collagenase (Gibco, Billings, MT, USA) in high-glucose Dulbecco’s modified Eagle’s medium (DMEM, Gibco) using a thermostatic shaker for 6 h at 37 °C. The isolated chondrocytes were then cultured using high-glucose DMEM supplemented with 10% fetal bovine serum (FBS, Gibco) and 1% antibiotics in a 5% CO_2_ incubator at 37 °C. Passaging was performed after 90% of the cells had fused.

RAW264.7 cells (murine macrophage cell lines) were obtained from the Type Culture Collection of the Chinese Academy of Sciences and cultured using the same method as was used for culturing the chondrocytes. All cells used in this study were second-generation.

### 2.4. Evaluation of the Cytocompatibility of the Cur/PLGA Nanofibrous Membrane

Chondrocyte suspensions were seeded onto the PLGA and Cur/PLGA nanofibrous membranes at a density of 1.0 × 10^5^ cells/mL. A chondrocyte suspension seeded into a Petri dish served as the control group. The chondrocyte-seeded samples were cultured in high-glucose DMEM supplemented with FBS and antibiotics for 7 days. Cell viability was assessed using the live/dead cell viability assay (Dojindo, Kumamoto, Japan) with a confocal laser scanning microscope (Leica, Wetzlar, Germany). The optical density (OD) values of the chondrocyte-seeded samples were obtained using the cell counting kit-8 (CCK-8) (Dojindo, Japan).

RAW264.7 macrophages were also seeded onto the PLGA and Cur/PLGA nanofibrous membranes at a density of 1.0 × 10^5^ cells/mL and cultured under the same condition as were used to culture the chondrocytes. After 24 h of culture, the samples were observed using scanning electron microscopy (SEM).

### 2.5. Evaluation of In Vitro Anti-Inflammatory Effect

RAW264.7 macrophages were pretreated with lipopolysaccharide (LPS; Gibco) (1 μg/mL) for 12 h and inoculated into the PLGA and Cur/PLGA nanofibrous membranes. The macrophage-loaded nanofibrous membranes were cultured in high-glucose DMEM supplemented with 5% FBS in a 5% CO_2_ incubator at 37 °C for 24 h. The samples from the two groups were subjected to Western blotting (WB) analysis and enzyme-linked immunosorbent assay (ELISA).

To perform WB analysis, total proteins were extracted from samples using radioimmunoprecipitation assay buffer (KeyGen, Changchun, China). Protein samples were resolved using sodium dodecyl sulfate–polyacrylamide gel electrophoresis. The resolved proteins were transferred to a polyvinylidene difluoride membrane. The membrane was blocked with 5% bovine serum albumin and probed with primary antibodies against inflammatory cytokines (IL-1β, IL-6, and TNF-A-α) (1:1000, Santa Cruz Biotechnology, Inc., Dallas, TX, USA). Next, the membrane was probed with horseradish-peroxidase-conjugated goat anti-rabbit/mouse IgG antibodies (1:10,000, Aspen Biotech, Shanghai, China). Immunoreactive signals were developed using the enhanced chemiluminescence kit (Aspen Biotech, Shanghai, China). The relative expression levels of proteins were quantified using Band-Scan software.

To perform ELISA, the culture supernatant was collected via centrifugation. The total protein concentration was quantified using the bicinchoninic acid method. The contents of IL-1β, IL-6, and TNF-A-α in the supernatant were measured using the ELISA kits (R&G Systems, North Port, FL, USA). After washing 6 times, the samples were incubated with the diluted standard product at 37 °C for 1 h. Next, the samples were washed and incubated with 50 μL of enzyme labeling reagent for 30 min. The samples were washed again and incubated with the developer and 3,3′,5,5′-tetramethylbenzidine terminator, which were added at 10–30 min intervals. The results were obtained in 15 min.

### 2.6. In Vitro Cartilage Regeneration

PLGA was homogeneously dissolved in trichloromethane solution at a concentration of 12% (*w*/*v*). The solution was poured into a customized cylindrical mold (diameter = 8 mm; height = 2 mm) and frozen at −20 °C for 12 h. After complete crystallization, the mold was lyophilized at −60 °C for 24 h to obtain a porous PLGA scaffold. The prepared chondrocyte suspension was seeded onto the porous PLGA scaffold at a concentration of 50 × 10^6^ cells/mL and subjected to stationary culture for 4 h in a 5% CO_2_ incubator at 37 °C. The chondrocyte-loaded PLGA was cultured using high-glucose DMEM supplemented with 5% FBS and 1% antibiotics in a 5% CO_2_ incubator at 37 °C. After 3 weeks, the obtained sample was imaged and subjected to histological analysis.

To perform histological examination, the samples were fixed in 4% paraformaldehyde, embedded in paraffin, and sectioned. The sections were stained with hematoxylin and eosin (H&E) and safranin-O to evaluate the structure and deposition of the cartilage extracellular matrix (ECM) in the engineered tissue, respectively. Immunohistochemical analysis of COL II was performed to verify the cartilage-specific phenotype, following a previously described method [24].

### 2.7. Subcutaneous Implantation

The PLGA and Cur/PLGA nanofibrous membranes were shaped into circular disks with a diameter of 12 mm. Next, the prepared in vitro engineered cartilage tissue was wrapped with two pieces of nanofibrous membrane. The edges of the nanofibrous membranes were tightly sealed via hot compression. Next, the membrane-packaged cartilage tissue was subcutaneously implanted into Sprague Dawley rats (n = 6 per group). The implanted samples were harvested after 2 and 4 weeks for gross, biochemical, biomechanical, and histological assessments.

The nanofibrous membranes were carefully stripped to expose the cartilage tissue. The samples were weighed using an electronic balance. Additionally, the samples were analyzed using a biomechanical analyzer (Instron-5542, Canton, MA, USA) with a continuous planar unconfined strain at a rate of 1 mm/min. The Young’s modulus value of the test samples was calculated based on the slope of the stress–strain curve.

The samples were digested in a papain solution (Sigma-Aldrich, St. Louis, MO, USA) at 65 °C. The sulfated glycosaminoglycan (GAG) contents were quantified using the Alcian blue method. The COL II contents in the samples were quantified using an ELISA kit (Nanjing Jiancheng, China).

The samples were also sectioned for histological evaluation, which included H&E staining to determine the cartilage structure, safranin-O and toluidine blue staining to assess the cartilage ECM, and COL II immunohistochemical staining to verify the cartilage-specific phenotype. Immunohistochemical staining of CD3 and CD68 was performed to evaluate immunocyte infiltration. Immunofluorescence staining of IL-1β, IL-6, and TNF-α was also performed. The captured images were analyzed using ImageJ software to calculate the relative staining intensity.

### 2.8. Statistical Analysis

All experiments were repeated independently at least three times. The results are expressed as the mean ± standard deviation. Means between multiple groups were compared using one-way analysis of variance, while those between two groups were compared using Student’s *t*-test. All statistical analyses were performed using GraphPad Prism 8 software. Differences were considered significant at *p* < 0.05.

## 3. Results

### 3.1. The Cur/PLGA Nanofibrous Membrane Was Successfully Developed with Sustained Release Kinetics and Satisfactory Stability of Cur

The Cur/PLGA nanofibrous membrane was prepared using the conventional electrospinning method. Pure PLGA solution exhibited a transparent appearance with a white color (Figure 1A1). The PLGA nanofibrous membrane also exhibited a white color (Figure 1A2). In contrast, the Cur/PLGA solution (Figure 1B1) and synthesized nanofibrous membrane (Figure 1B2) supplemented with Cur exhibited a yellow color. The micromorphology evaluated using SEM indicated that both the PLGA (Figure 1A3,A4) and Cur/PLGA (Figure 1B3,B4) nanofibrous membranes exhibited similar nanoscale fibers and porous structures, and the porosity of both the PLGA and Cur/PLGA nanofibrous membranes was also similar, at 74.26% in PLGA and 76.13% in Cur/PLGA.

The FTIR spectrum revealed (Figure 1C) that the PLGA sample exhibited a typical peak at 1752 cm^−1^ (corresponding to C=O) and a relatively weak absorption peak at 1382 cm^−1^. The Cur sample exhibited a typical peak at 1509 cm^−1^ (corresponding to C=C). The Cur/PLGA sample exhibited characteristic peaks of both pure PLGA and Cur samples. This indicated the successful development of the Cur/PLGA nanofibrous membranes. The Cur/PLGA sample exhibited new absorption peaks at 1600 and 1585 cm^−1^, which may be attributed to the deformation and vibration of the benzene ring skeleton in the Cur structure. HPLC analysis revealed the typical peaks of Cur in the Cur/PLGA sample (Figure 1D), further confirming the successful encapsulation of Cur into PLGA.

The results of the in vitro release kinetics assay revealed the sustained release of Cur from the Cur/PLGA nanofibrous membranes over 14 days in vitro upon incubation in PBS (Figure 1E). In particular, a relative burst release of Cur occurred in the initial 4 days. Subsequently, a slow release of Cur was observed on days 12–14. The stability of Cur in the Cur/PLGA nanofibrous membranes was further evaluated. After incubation in PBS, almost 80% of pure Cur underwent rapid degradation after 8 h. However, 90% of Cur in the Cur/PLGA samples did not undergo degradation after 8 h (Figure 1F). These data indicated satisfactory stability of the loaded Cur in the Cur/PLGA nanofibrous membranes.

### 3.2. The Synthesized Cur/PLGA Nanofibrous Membrane Is Cytocompatible

To assess the cytocompatibility of the Cur/PLGA nanofibrous membrane, both chondrocytes and RAW264.7 macrophages were seeded onto the PLGA and Cur/PLGA nanofibrous membranes. Live/dead staining revealed that the viability rates of chondrocytes in both the PLGA and Cur/PLGA groups were similar, with comparable adhesion and proliferation rates, but lower than those of chondrocytes in the control (Petri dish) group (Figure 2A), as evidenced by the increased numbers of green-stained live cells. The number of dead cells was low in all three groups, as indicated by the lack of red-stained cells. The quantitative analysis of OD values determined using the CCK-8 assay further confirmed the comparable viability rates in the PLGA and Cur/PLGA groups and the increased viability rates in the control group (Figure 2B). Additionally, SEM analysis revealed that the RAW264.7 macrophages could adhere to the surface of both PLGA and Cur/PLGA nanofibrous membranes (Figure 2C). The size of RAW264.7 macrophages was larger than that of the pores of both the PLGA and Cur/PLGA nanofibrous membranes.

### 3.3. The Cur/PLGA Nanofibrous Membrane Exerted a Satisfactory Anti-Inflammatory Effect In Vitro

To investigate the anti-inflammatory effect of the Cur/PLGA nanofibrous membrane, LPS-pretreated RAW264.7 macrophages were incubated with both PLGA and Cur/PLGA nanofibrous membranes for 24 h. WB analysis suggested that the expression levels of inflammatory cytokines (IL-1β, IL-6, and TNF-α) in the Cur/PLGA group were lower than those in the PLGA group (Figure 3A). Quantitative analysis of blots further confirmed that the expression levels of IL-1β, IL-6, and TNF-α in the Cur/PLGA group were significantly lower than those in the PLGA group (Figure 3B). Consistently, the results of ELISA demonstrated that the expression levels of IL-1β, IL-6, and TNF-α in the Cur/PLGA group were lower than those in the PLGA group (Figure 3B). These results indicate that the addition of Cur imparted the Cur/PLGA nanofibrous membrane with anti-inflammatory properties.

### 3.4. The Successful Generation of Cartilage Tissue In Vitro

A porous PLGA scaffold was fabricated using the conventional freeze-drying method (Figure 4A). SEM analysis revealed that the PLGA scaffold exhibited a porous structure with a pore size of approximately 200 μm (Figure 4B), which was conducive to chondrocyte adhesion and infiltration. The porous PLGA scaffold was seeded with chondrocytes and subjected to in vitro culture for 3 weeks. The gross observation of the generated sample indicated a typical cartilage appearance with a smooth texture and ivory-white color (Figure 4C). Histological evaluation of the generated sample using H&E staining revealed a preliminary cartilage structure with a lacunar structure (Figure 4D). Meanwhile, safranin-O staining revealed abundant GAG deposition (Figure 4E). Immunohistochemical staining confirmed enhanced COL II secretion (Figure 4F). These results verified the successful generation of cartilage tissues in vitro.

### 3.5. Enhanced Cartilage Regeneration via Subcutaneous Implantation of Nanofibrous-Membrane-Packaged Cartilage Tissue

As shown in Figure 5A, the prepared PLGA and Cur/PLGA nanofibrous membranes were used to tightly pack the in vitro generated cartilage tissue. The samples were then subcutaneously implanted into rats. In the control group, an in vitro generated cartilage tissue that was not packaged with a nanofibrous membrane was subcutaneously implanted into rats (Figure 5B). At weeks 2 and 4 post-implantation, the Cur/PLGA nanofibrous-membrane-packaged samples exhibited a white color, whereas the PLGA nanofibrous-membrane-packaged samples exhibited an appearance consistent with inflammation, with a red color (Figure 5C). After the overlapping nanofibrous membranes were stripped, the exposed samples in the Cur/PLGA group exhibited a typical cartilage-like appearance with a smooth texture and ivory-white color, whereas the samples in the PLGA and control groups exhibited a fibrosis-like appearance with a red color (Figure 5D). Fibrosis tendencies progressed as the implantation period increased from 2 weeks to 4 weeks. Meanwhile, fibrosis tendencies in the control group were aggravated when compared with those in the PLGA group. The ranking of wet weight, Young’s modulus, GAG content, and COL II content was as follows: Cur/PLGA group > PLGA group > control group. These values decreased as the implantation period increased (from 2 weeks to 4 weeks) (Figure 5E).

Histological analysis was performed to determine the real microstructure and tissue compositions of the in vivo generated samples. At weeks 2 and 4, the embedded cartilage samples in the Cur/PLGA group exhibited a comprehensive cartilage-specific ECM deposition when compared with those in the PLGA and control groups, as evidenced by the detection of a typical lacunar structure using H&E staining (Figure 6A1–F1), strongly positive GAG secretion using safranin-O (Figure 6A2–F2) and toluidine blue staining (Figure 6A3–F3), and upregulated COL II deposition using immunohistochemical staining (Figure 6A4–F4). Additionally, the cartilaginous phenotype decreased with the increase in the length of implantation period in all three groups. The cartilaginous phenotype was almost completely lost in both the PLGA and control groups at week 4. These data suggest that the cartilage packaged with the Cur/PLGA nanofibrous membrane significantly retained the cartilaginous phenotype in immunocompetent animals.

### 3.6. The Cur/PLGA Nanofibrous Membrane Exhibited a Favorable Anti-Inflammatory Effect In Vivo

Next, the mechanisms of the Cur/PLGA nanofibrous membrane in maintaining the cartilaginous phenotype in immunocompetent animals were examined. The inflammatory cells (CD3-positive and CD68-positive cells) were examined using immunohistochemical staining. Based on the attenuated positive staining in the Cur/PLGA group and increased positive staining in the PLGA and control groups for both CD3 (Figure 7A1–F1) and CD68 (Figure 7A2–F2), the degree of infiltration of inflammatory cells at both 2 and 4 weeks was as follows: Cur/PLGA group < PLGA group < control group. Additionally, the expression patterns of the inflammatory cytokines IL-1β (Figure 7A3–F3), IL-6 (Figure 7A4–F4), and TNF-α (Figure 7A5–F5) exhibited the trend of Cur/PLGA group < PLGA group < control group at both week 2 and week 4, as evidenced by decreased immunofluorescence staining intensities in the Cur/PLGA group and upregulated immunofluorescence staining intensities in the control group. The positive staining intensities of CD3, CD68, IL-1β, IL-6, and TNF-α at week 4 were higher than those at week 2 in the Cur/PLGA, PLGA, and control groups. The positive staining intensities of CD3, CD68, IL-1β, IL-6, and TNF-α in the cartilage region were lower than those outside the nanofibrous membrane in both the Cur/PLGA and PLGA groups. Moreover, the quantification of CD3 and CD68 immunohistochemical staining intensities and IL-1β, IL-6, and TNF-α immunofluorescence staining intensities (Figure 7G–K) indicated the attenuation of inflammatory responses in the Cur/PLGA group and the upregulation of inflammatory responses in the control group relative to the PLGA group with the increase in the length of the implantation period from 2 weeks to 4 weeks.

## 4. Discussion

Tissue engineering methods have been successfully used to generate cartilage with a tunable shape in vitro using several kinds of scaffolds, such as porous scaffolds [25], nanofilms [26], or hydrogels [27]. However, the implantation of in vitro engineered cartilage into immunocompetent animals promotes immune responses against the cartilage tissue, leading to the erosion and resorption of the cartilage’s ECM [28]. The following three mechanisms are proposed to underlie the immune response against implanted cartilage tissue: (1) the direct contact of the biomaterials with the immune system; (2) the in vitro cultured cartilage may lose its original phenotype after multiple passages, leading to the recognition of autologous cells as foreign cells by the immune system; (3) the surgical procedure for the implantation of in vitro engineered cartilage leads to trauma, resulting in an inflammatory reaction. To promote cartilage regeneration in vivo and advance the clinical translation of in vitro engineered cartilage, optimal strategies must be devised to suppress immune responses.

The healthy cartilage tissue has an ECM with a dense structure [23]. Hence, mature cartilage exhibits an enhanced capacity to resist inflammatory cell infiltration. Based on this theory, a previous study proposed that in vitro pre-cultivation can alleviate post-implantation inflammation and enhance the development of tissue-engineered tubular cartilage [28]. However, several studies have indicated that the in vitro engineered cartilage is immature, with preliminary cartilage ECM (which was also confirmed in this study (Figure 4)). Although prolonged in vitro cultivation can improve the maturity of cartilage to a certain extent, the biomechanical properties and biochemical composition of the in vitro engineered cartilage are inferior to those of healthy mature cartilage tissues [29,30]. In vivo implantation significantly enhances the maturity of the implanted cartilage [22,31]. Thus, in vitro engineered cartilage can be considered healthy and effective to restore cartilage defects only when the inflammatory response against it is suppressed.

Several studies have reported satisfactory and robust cartilage regeneration in nude mice [32,33], which can be mainly attributed to the immunocompromised status of the animals, which protects the implanted cartilage against immune attack. Consequently, generating an immunosuppressive niche in immunocompetent animals is a good strategy to accommodate the in vitro engineered cartilage. Inflammatory reactions mainly involve inflammatory cells and cytokines [34]. Hence, an ideal immunosuppressive niche should be effective in preventing the infiltration of both inflammatory cells and cytokines.

Fish are trapped using fishing nets, which is based on the principle that the size of the fish is larger than the pore size of the fishing net. Based on the working principle of fishing nets, the development of fishing-net-like nanomaterials can impede the infiltration of immunocytes. A recent study used a similar principle to block the infiltration of vascular endothelial cells [35], indicating the feasibility of this strategy. However, the inflammatory reaction involves both inflammatory cells and cytokines. Thus, the physical inhibition of immunocyte infiltration may not completely protect the in vitro engineered cartilage against inflammatory responses. The most effective approach to suppress inflammatory cytokines is the usage of anti-inflammatory drugs. Hence, the development of a fishing-net-like material with a small pore size and loaded with an anti-inflammatory drug is desirable. In the present study, an electrospun nanofibrous membrane was employed as a semipermeable membrane to block the infiltration of inflammatory cells, and Cur was loaded into the nanofibrous membrane to endow it with an antagonistic effect against inflammatory cytokines.

The usage of the Cur/PLGA nanofibrous membrane as a physical barrier against inflammatory cells has the following advantages: (1) The preparation of the Cur/PLGA nanofibrous membrane is simple and cost-effective. (2) The porous structure of the nanofibrous membrane does not affect the exchange of nutrients between the inner cartilage and the external space. (3) The nanofibrous membrane exhibits strong plasticity and can fit well in tissues with complex shapes. Thus, the membrane can be used to wrap the cartilages of different shapes associated with different parts, such as the ear, nose, and trachea. (4) As blood cell infiltration is reported to promote ossification, the Cur/PLGA nanofibrous membrane can also function as a physical barrier for vascular endothelial cells.

The pure PLGA nanofibrous membrane could not inhibit inflammatory cell infiltration, which was suspected to be due to the inability of the pure PLGA nanofibrous membrane to suppress inflammatory cytokines. Hence, Cur—an anti-inflammatory drug—was added to the PLGA membrane to obtain the Cur/PLGA nanofibrous membrane. Although Cur has several drawbacks (e.g., instability, poor solubility, and low bioavailability) [36], the roles of Cur in suppressing inflammatory cytokines have been extensively studied [37,38]. Cur exerts anti-inflammatory activities through the following mechanisms: (1) Cur alleviates oxidative stress and inflammation in chronic diseases by modulating the Nrf2-keap1 signaling pathway. (2) Cur can suppress pro-inflammatory pathways associated with most chronic conditions and inhibit both the production of TNF-α and the TNF-α-mediated cell signaling in different cell types [39]. (3) The specific chemical structure of Cur may enable it to function as a natural free-radical scavenger, which can reduce the release of various interleukins via NF-κB [40]. This study demonstrated that the Cur/PLGA nanofibrous membrane exerted a satisfactory anti-inflammatory effect both in vitro and in vivo.

The FTIR spectrum confirmed the successful loading of Cur into the PLGA nanofibrous membrane. HPLC analysis demonstrated that the Cur/PLGA samples exhibited identical typical peaks to those observed in the pure Cur sample, suggesting that the electrospinning procedure does not alter the chemical structure of Cur and, consequently, maintains the original anti-inflammatory effect of Cur. Additionally, the in vitro release curve suggested that the release of Cur from the Cur/PLGA nanofibrous membrane involved the following three distinct stages: (1) Slow release kinetics in the initial 1–4 days, which may be attributed to the slow degradation of the PLGA polymer. (2) Accelerated release kinetics in the intermediate 4–12 days. The gradual degradation of the PLGA polymer significantly increased the contact area with PBS, which further promoted the degradation of the PLGA polymer and the release of Cur. (3) Decelerated release kinetics in the final 12–14 days, which may be attributed to the insufficient content of remaining Cur.

This study successfully established an immunosuppressive niche by tightly packaging the in vitro engineered cartilage with the Cur/PLGA nanofibrous membrane, which significantly promoted cartilage regeneration in immunocompetent animals. However, several limitations must be addressed before the clinical application of this engineered cartilage. First, Cur was almost completely released by day 14, which is a non-optimal time course for cartilage maturation (as evidenced by the mild immune responses against the regenerated cartilage in the Cur/PLGA group). Future studies must develop a new system to achieve prolonged release of Cur. Additionally, only one concertation of Cur was investigated in this study. Efforts are ongoing to select the optimal Cur concentration for practical utilization. Furthermore, the maximum in vivo observation period in this study was only 4 weeks. Long-term observations, such as 1-year observations, are needed. Finally, this study only used a rat model. The efficacy of the model developed in this study must be evaluated in other animals with complex immune systems, such as sheep or swine.

## 5. Conclusions

This study successfully developed a Cur/PLGA nanofibrous membrane with nanoscale pore size and anti-inflammatory activities. The Cur/PLGA nanofibrous membrane exhibited sustained Cur release kinetics and enhanced Cur stability. Additionally, the expression of inflammatory cytokines in the Cur/PLGA nanofibrous-membrane-implanted group was significantly downregulated when compared with that in the PLGA-implanted group. Moreover, the packaged Cur/PLGA nanofibrous membrane effectively generated an immunosuppressive niche, protecting the cartilage against inflammatory cell infiltration after in vivo implantation. The approach used in this study can potentially enable the clinical translation of tissue-engineered cartilage to restore cartilage defects.

## Figures and Tables

**Figure 1 membranes-13-00335-f001:**
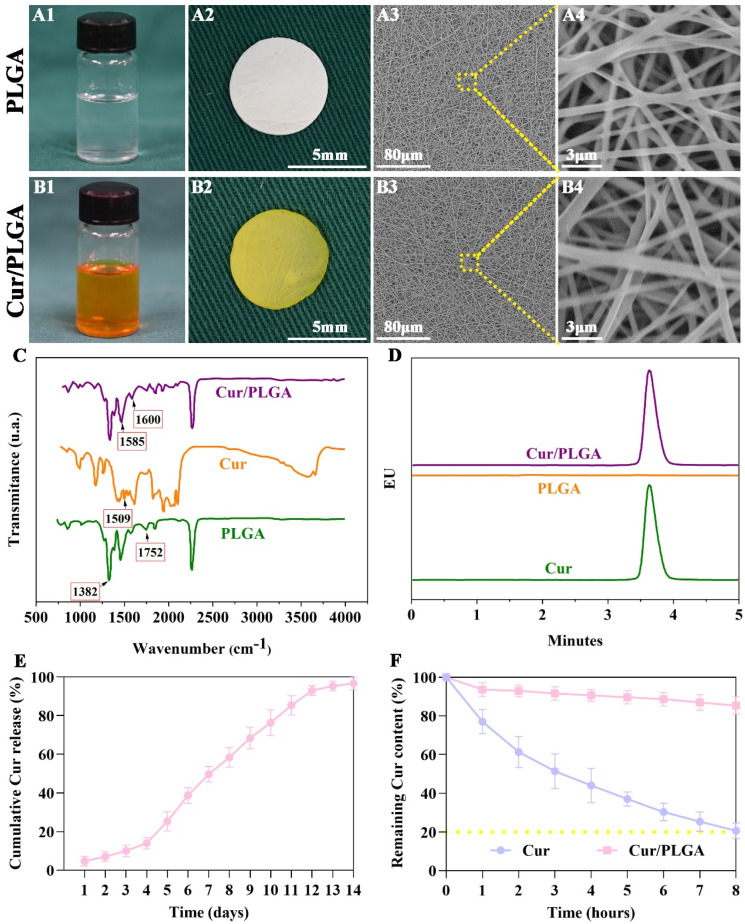
Synthesis and characterization of the poly(lactic−co−glycolic acid) (PLGA) and curcumin/PLGA (Cur/PLGA) nanofibrous membranes: Images of spinning solutions of the PLGA (**A1**) and Cur/PLGA (**B1**) groups. Images of fabricated nanofibrous membranes of the PLGA (**A2**) and Cur/PLGA (**B2**) groups. Scanning electron micrographs of nanofibrous membranes of the PLGA (**A3**,**A4**) and Cur/PLGA (**B3**,**B4**) groups. Fourier−transform infrared (FTIR) spectra (**C**) and high−performance liquid chromatography chromatograms (**D**) of Cur, PLGA, and Cur/PLGA samples. Curve representing the cumulative release of Cur from the Cur/PLGA nanofibrous membrane (**E**). Remaining Cur content in the pure Cur and Cur/PLGA groups after incubation with phosphate−buffered saline (PBS) (37 °C, pH 7.4) for 8 h (**F**).

**Figure 2 membranes-13-00335-f002:**
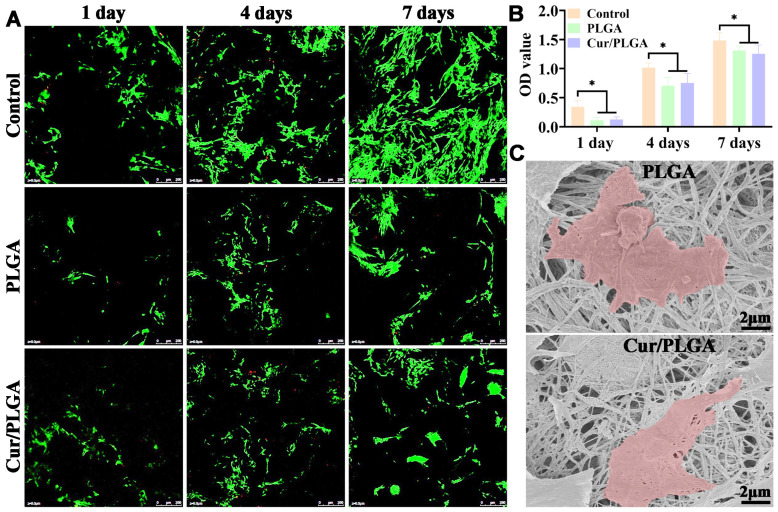
In vitro cytocompatibility of the curcumin/poly(lactic-co-glycolic acid) (Cur/PLGA) nanofibrous membrane: Live/dead staining (**A**) and optical density (OD) values, which were determined using the cell counting kit (CCK)-8 assay (**B**), of chondrocytes seeded onto the Petri dish (control), PLGA, and Cur/PLGA nanofibrous membranes for 1–7 days. Scanning electron microscopy analysis of RAW264.7 macrophages seeded onto the PLGA and Cur/PLGA nanofibrous membranes for 24 h (**C**). The red color indicates the RAW264.7 macrophages. * *p* < 0.05.

**Figure 3 membranes-13-00335-f003:**
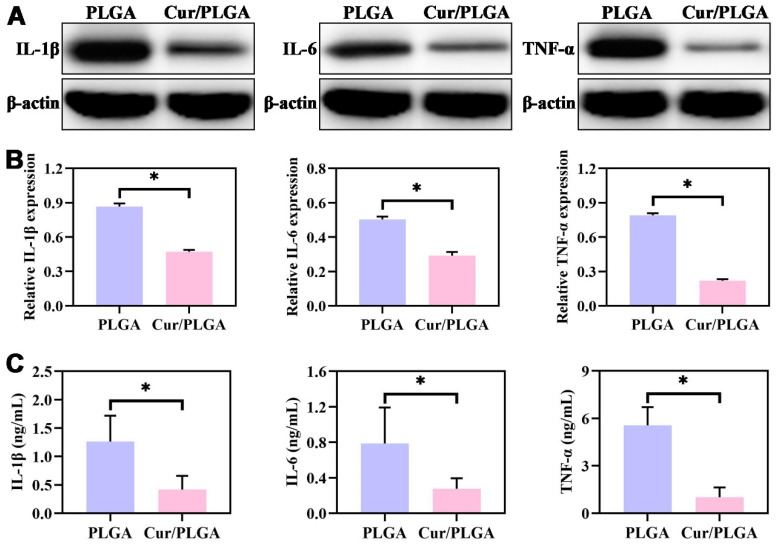
In vitro anti-inflammatory activities of the curcumin/poly(lactic-co-glycolic acid) (Cur/PLGA) nanofibrous membrane: The expression levels of IL-1β, IL-6, and TNF-α in the PLGA and Cur/PLGA nanofibrous membranes incubated with RAW264.7 macrophages for 24 h were determined using Western blotting (**A**). The quantification of the expression levels of IL-1β, IL-6, and TNF-α in the PLGA and Cur/PLGA groups from the blots (**B**). The expression levels of IL-1β, IL-6, and TNF-α in the PLGA and Cur/PLGA nanofibrous membranes incubated with RAW264.7 macrophages for 24 h were also examined using an enzyme-linked immunosorbent assay (**C**). * *p* < 0.05.

**Figure 4 membranes-13-00335-f004:**
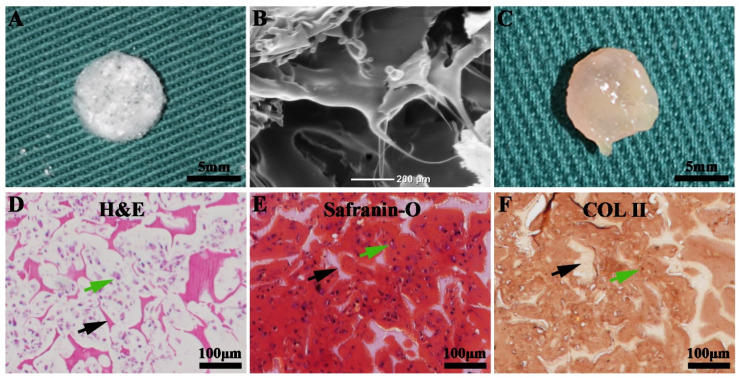
In vitro cartilage regeneration: Images (**A**) and scanning electron micrographs (**B**) of a porous poly(lactic-co-glycolic acid) (PLGA) scaffold. Images (**C**), hematoxylin-and-eosin staining (**D**), safranin-O staining (**E**), and COL II immunohistochemical staining (**F**) of the PLGA scaffold seeded with chondrocytes for 3 weeks. The green arrows indicate the cartilage-specific extracellular matrix (ECM). The black arrows indicate the PLGA scaffold.

**Figure 5 membranes-13-00335-f005:**
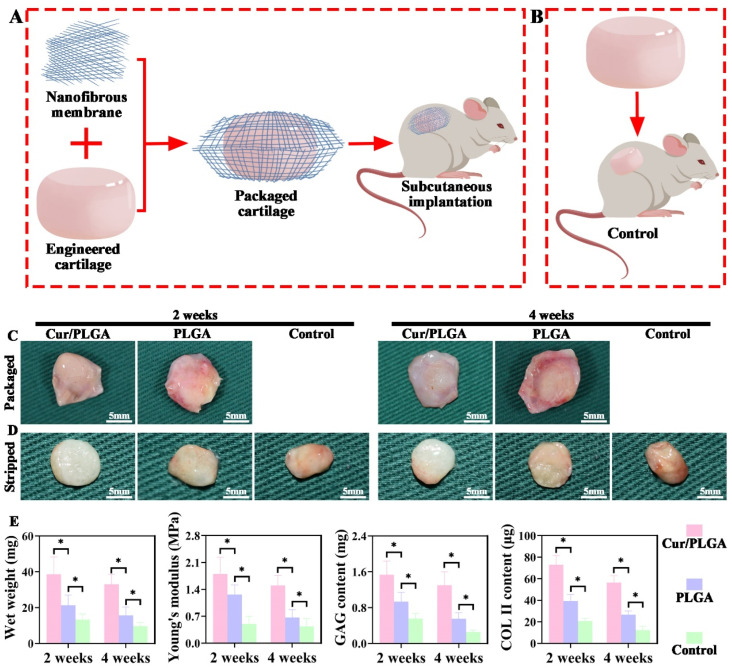
Subcutaneous implantation of nanofibrous-membrane-packaged cartilage tissues for 2 and 4 weeks: The schematic illustration of the preparation and subcutaneous implantation of nanofibrous-membrane-packaged cartilage tissue in rats (**A**) and the corresponding control groups (**B**). Images of the nanofibrous-membrane-packaged cartilage tissue in the curcumin/poly(lactic-co-glycolic acid) (Cur/PLGA) and PLGA groups (**C**). Images of the nanofibrous-membrane-stripped cartilage tissue in the Cur/PLGA and PLGA groups and the corresponding control group (**D**). Quantitative analysis of wet weight, Young’s modulus, glycosaminoglycan (GAG) content, and COL II content in the nanofibrous-membrane-stripped cartilage tissue of the Cur/PLGA, PLGA, and control groups (**E**). * *p* < 0.05.

**Figure 6 membranes-13-00335-f006:**
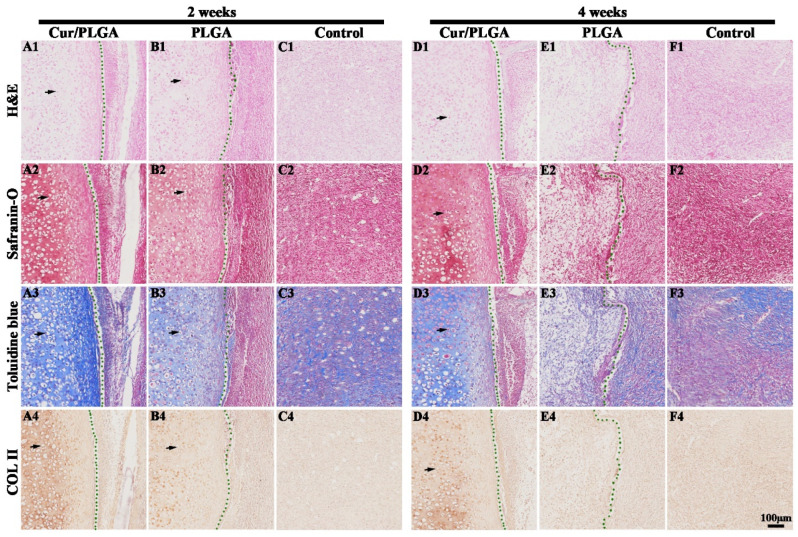
Histological evaluation of the nanofibrous-membrane-packaged cartilage tissue at weeks 2 and 4 after subcutaneous implantation in rats: Hematoxylin and eosin (H&E) staining of samples in the curcumin/poly(lactic-co-glycolic acid) (Cur/PLGA), PLGA, and control groups (**A1**–**F1**). Safranin-O staining of samples in the Cur/PLGA, PLGA, and control groups (**A2**–**F2**). Toluidine blue staining of samples in the Cur/PLGA, PLGA, and control groups (**A3**–**F3**). COL II immunohistochemical staining of samples in the Cur/PLGA, PLGA, and control groups (**A4**–**F4**). The black arrows indicate the cartilage-specific extracellular matrix (ECM). The dotted green lines outline the nanofibrous membrane, while the remaining region represents the cartilage.

**Figure 7 membranes-13-00335-f007:**
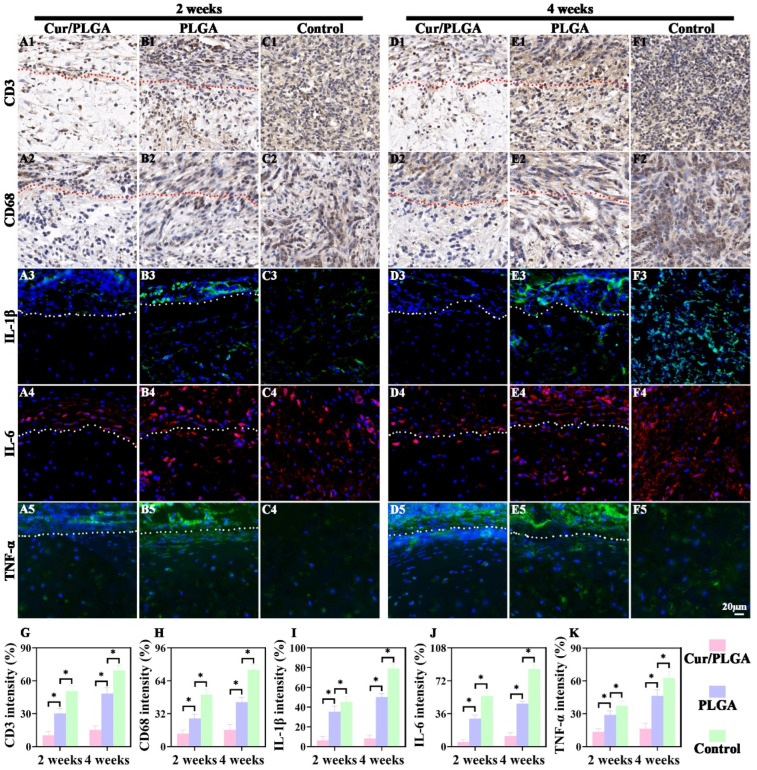
Anti-inflammatory effects of the curcumin/poly(lactic-co-glycolic acid) (Cur/PLGA) nanofibrous membrane at weeks 2 and 4 after subcutaneous implantation in rats: CD3 immunohistochemical staining of samples in the Cur/PLGA, PLGA, and control groups (**A1**–**F1**). CD68 immunohistochemical staining of samples in the Cur/PLGA, PLGA, and control groups (**A2**–**F2**). IL-1β immunofluorescence staining of samples in the Cur/PLGA, PLGA, and control groups (**A3**–**F3**). IL-6 immunofluorescence staining of samples in the Cur/PLGA, PLGA, and control groups (**A4**–**F4**). TNF-α immunofluorescence staining of samples in the Cur/PLGA, PLGA, and control groups (**A5**–**F5**). Relative staining intensities of CD3, CD68, IL-1β, IL-6, and TNF-α in the Cur/PLGA, PLGA, and control groups (**G**–**K**). The dotted red and white lines outline the nanofibrous membrane, and the lower region represents the cartilage. * *p* < 0.05.

## Data Availability

The data that support the findings of this study are available from the corresponding author upon reasonable request.

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
