# Peer review of "The Immunosuppressive Niche Established with a Curcumin-Loaded Electrospun Nanofibrous Membrane Promotes Cartilage Regeneration in Immunocompetent Animals"

_membranes, 2023, doi:10.3390/membranes13030335_

Round 1
Reviewer 1 Report
In general, this manuscript on "the immunosuppressive niche established with curcumin-loaded electrospun nanofibrous membrane promotes cartilage regeneration in an immunocompetent animal" was well presented except for the discussion part -- see page 12, line 394 - page 13, line 442 - the information in the section is more suitable for the introduction/literature review. Consider relocating this part accordingly.
The phrase subheadings for the results section should be changed to sound like results, not repeating the research methods.
So, for the discussion part -- to arrange the contents (for each finding/answer to the research question) following the typical discussion elements below:
(1) HIGHLIGHT the findings of the study (but not repeating results reporting, because the results already had a specific section for it)
(2) COMPARE and CONTRAST the findings with other studies (find articles that support or are in line with the present findings and those against the present findings)
(3) JUSTIFY the present findings and suggest (or give reasons) why it ‘resulted’ as such; give/cite articles to support the justifications
Author Response
Detailed Responses to the Reviewer 1
In general, this manuscript on "the immunosuppressive niche established with curcumin-loaded electrospun nanofibrous membrane promotes cartilage regeneration in an immunocompetent animal" was well presented except for the discussion part -- see page 12, line 394 - page 13, line 442 - the information in the section is more suitable for the introduction/literature review. Consider relocating this part accordingly.
Response: Thank you for your positive and kindly comments. We have amended our work according to the reviewer’s useful suggestion, thus resulting in a considerable improvement of our manuscript.
Additionally, we have moved the part of page 12, line 394 - page 13, line 442 to the introduction. Please note the revised text was highlighted in red.
The phrase subheadings for the results section should be changed to sound like results, not repeating the research methods.
Response: Thank you for your kindly suggestion. We have rephrased the subheadings for the results section with sound like results. Please note the revised text was highlighted in red.
So, for the discussion part -- to arrange the contents (for each finding/answer to the research question) following the typical discussion elements below:
(1) HIGHLIGHT the findings of the study (but not repeating results reporting, because the results already had a specific section for it)
(2) COMPARE and CONTRAST the findings with other studies (find articles that support or are in line with the present findings and those against the present findings)
(3) JUSTIFY the present findings and suggest (or give reasons) why it ‘resulted’ as such; give/cite articles to support the justifications
Response: Thank you for your professional and useful comments. We have rewritten our discussion part according to the typical discussion elements. Please note the revised text was highlighted in red.

Reviewer 2 Report
CConsidering the complexity of the problem and the number of publications on this subject, I find the introduction embarrassingly modest.There is no broader reference here to the different methods of cartilage regeneration. Was this literature not known to the authors? And only two references on the properties of curcumin are probably not enough! Particularly considering the conclusions of DOI: 10.1021/acsmedchemlett.7b00139 An extension of the introduction is necessary.
At the same time, I believe that a significant part of the Discussion paragraph should be included in the Introduction.
There is no information whether the SEM micrographs were made without or with sputtering. And if sputtering was used, what guide was used. This is key information to evaluate micrographs. Also no magnifications or scale marks are given (Fig.1). All this should be completed.
Has membrane porosity been assessed by SEM alone?
And what is the cut off of non-woven fabric?
Fig. 4: Pictures A and C; Fig. 5 C and D: What is the scale unit?
Author Response
Detailed Responses to the Reviewer 2
Considering the complexity of the problem and the number of publications on this subject, I find the introduction embarrassingly modest. There is no broader reference here to the different methods of cartilage regeneration. Was this literature not known to the authors? And only two references on the properties of curcumin are probably not enough! Particularly considering the conclusions of DOI: 10.1021/acsmedchemlett.7b00139. An extension of the introduction is necessary.
Response: Thank you for your positive and kindly comments. We have amended our work according to the reviewer’s useful suggestion, thus resulting in a considerable improvement of our manuscript.
We have added some other references using the different methods for cartilage regeneration in our revised manuscript, including Ref. 32 and 33.
Additionally, we have also added some other references on the properties of curcumin, including Ref. 20, 36 and 37.
Moreover, we also referred to conclusions (Line 469-470) in DOI: 10.1021/acsmedchemlett.7b00139 as Ref. 36.
Please note the revised text was highlighted in red.
At the same time, I believe that a significant part of the Discussion paragraph should be included in the Introduction.
Response: Thank you for your kindly suggestion. We have moved the Discussion paragraph of page 12, line 394 - page 13, line 442 to the introduction. Please note the revised text was highlighted in red.
There is no information whether the SEM micrographs were made without or with sputtering. And if sputtering was used, what guide was used. This is key information to evaluate micrographs. Also no magnifications or scale marks are given (Fig.1). All this should be completed.
Response: Thank you for your careful observations. The SEM micrographs were made with sputtering using platinum (10 nm). We have added this information in our revised version. Please note the revised text was highlighted in red.
In addition, we have added scale marks in the SEM micrographs and their corresponding magnifications in Fig. 1.
Has membrane porosity been assessed by SEM alone?
Response: Thank you for your comments. We have added the assessment of membrane porosity by SEM images using a Photoshop 2021 software. The porosity of PLGA nanofibrous membrane is 74.26%, and the Cur/PLGA nanofibrous membrane is 76.13%. And the corresponding results were added in our revised version.
And what is the cut off of non-woven fabric?
Response: Thank you for your comments. The non-woven fabric of nanofibrous membrane was cut off using a surgical blade.
Fig. 4: Pictures A and C; Fig. 5 C and D: What is the scale unit?
Response: Thank you for careful observation. We added the unit in Fig. 4: Pictures A and C; Fig. 5 C and D of our revised version.

Round 2
Reviewer 2 Report
Cut off for membranes is a value that determines the size of particles (chemicals, cells, pyrogens, etc.). It has nothing to do with the method of cutting the fleece! Nanofibers are semi-permeable membranes. When conducting research with semi-permeable membranes, it is worth knowing the elementary concepts of this field!